# Virtual Reality as a Tool for Upper Limb Rehabilitation in Rett Syndrome: Reducing Stereotypies and Improving Motor Skills

**DOI:** 10.3390/pediatric17020049

**Published:** 2025-04-18

**Authors:** Rosa Angela Fabio, Martina Semino, Michela Perina, Matteo Martini, Emanuela Riccio, Giulia Pili, Danilo Pani, Manuela Chessa

**Affiliations:** 1Department of Biomedical, Dental and Morphological and Functional Imaging Sciences, University of Messina, 98100 Messina, Italy; 2Airett Innovation and Research Center, 37100 Verona, Italy; martina.semino@airett.it (M.S.); michela.perina@airett.it (M.P.); emanuela.riccio@airett.it (E.R.); 3TICE Cooperativa Sociale, Via De Meis 25, 29121 Piacenza, Italy; 4Department of Informatics, Bioengineering, Robotics, and Systems Engineering, University of Genoa, 16100 Genoa, Italy; matteo.martini@edu.unige.it (M.M.); manuela.chessa@unige.it (M.C.); 5Department of Electrical and Electronic Engineering, University of Cagliari, 09121 Cagliari, Italydanilo.pani@unica.it (D.P.)

**Keywords:** Rett Syndrome, virtual reality, rehabilitation, upper limb motor skills, stereotypies, motor abilities

## Abstract

Background/Objectives: Rett Syndrome (RTT) is a rare neurodevelopmental disorder that causes the loss of motor, communicative, and cognitive skills. While no cure exists, rehabilitation plays a crucial role in improving quality of life. Virtual Reality (VR) has shown promise in enhancing motor function and reducing stereotypic behaviors in RTT. This study aims to assess the impact of VR training on upper limb motor skills in RTT patients, focusing on reaching and hand-opening tasks, as well as examining its role in motivation and engagement during rehabilitation. Methods: Twenty RTT patients (aged 5–33) were randomly assigned to an experimental group (VR training) and a control group (standard rehabilitation). Pre- and post-tests evaluated motor skills and motivation in both VR and real-world contexts. The VR training involved 40 sessions over 8 weeks, focusing on fine motor tasks. Non-parametric statistical methods were used to analyze the data. Results: Results indicated significant improvements in the experimental group for motor parameters, including reduced stereotypy intensity and frequency, faster response times, and increased correct performance. These improvements were consistent across VR and ecological conditions. Moreover, attention time increased, while the number of aids required decreased, highlighting enhanced engagement and independence. However, motivation levels remained stable throughout the sessions. Conclusions: This study demonstrates the potential of VR as a tool for RTT rehabilitation, addressing both motor and engagement challenges. Future research should explore the customization of VR environments to maximize the generalization of skills and sustain motivation over extended training periods.

## 1. Introduction

Rett Syndrome (RTT) is a rare genetic neurodevelopmental disorder predominantly affecting females, with an incidence of approximately 1 in 10,000 live-born girls [1,2]. This condition, caused primarily by mutations in the MECP2 gene, presents a diverse clinical spectrum that makes diagnosis and treatment particularly challenging. Classic RTT is characterized by a loss of previously acquired skills, including gross motor abilities, purposeful hand movements, and verbal communication, often accompanied by stereotypic hand motions such as wringing or tapping [3]. Additional hallmark features include slowed head growth leading to acquired microcephaly, intellectual disabilities, and significant challenges with motor coordination, classified as apraxia or dyspraxia [4].

Beyond these primary features, RTT is often associated with a range of comorbidities, including musculoskeletal abnormalities such as scoliosis and foot deformities, autonomic irregularities like hyperventilation and apnea, and gastrointestinal dysfunctions including chronic constipation and gastroesophageal reflux. Mood instability and episodes of self-injurious behaviors further complicate care for individuals with RTT [5].

Although no cure exists to date, targeted pharmacological interventions have proven beneficial in managing symptoms. For instance, medications can help control seizures, regulate respiratory and cardiac functions, and address sleep and behavioral disturbances [4,6,7]. Rehabilitation, however, plays an irreplaceable role in sustaining functional abilities, mitigating deterioration, and enhancing quality of life for individuals with RTT and their caregivers. Telerehabilitation has also proven effective in addressing RTT-specific challenges, particularly during the COVID-19 pandemic. Caprì et al. [8] proposed a holistic telerehabilitation model for RTT, emphasizing its feasibility for remote motor and cognitive training. Combining this approach with VR technologies could further enhance accessibility and effectiveness, especially for individuals with limited mobility or those in remote locations.

Tailored rehabilitation programs, incorporating frequent but low-intensity activities, have been shown to improve developmental outcomes, particularly when designed with the active involvement of families and conducted across both home and community settings [9,10].

In practice, rehabilitation for RTT often involves an interdisciplinary approach. Physical and occupational therapists work collaboratively to improve gross and fine motor skills, mitigate stereotypical behaviors, and promote engagement in meaningful activities [5]. However, dyspraxia and stereotypical movements typical of RTT often limit the effectiveness of traditional rehabilitation methods. These limitations underscore the need to explore innovative, engaging, and adaptive rehabilitation strategies tailored to this population.

### 1.1. Virtual Reality for Rehabilitation

Virtual Reality (VR) has emerged as a transformative tool in rehabilitation since the 1990s, offering immersive and interactive environments that simulate real-world tasks. VR technology provides a unique platform for engaging patients in therapeutic activities, enhancing motivation and participation—critical factors in successful rehabilitation, especially for individuals with neurodevelopmental disorders [10,11]. For example, VR has been shown to facilitate motor and cognitive skill acquisition within controlled, low-risk settings where patients can safely practice tasks without the fear of real-world consequences [12,13,14].

Recent advancements have expanded the accessibility of VR, making it a viable option for integrating into rehabilitation programs. For individuals with intellectual and neurodevelopmental disabilities, immersive VR has demonstrated superior outcomes compared to non-immersive devices in fostering real-world skill development [15]. Immersive and non-immersive media differ regarding the user’s perspective and the level of engagement they provide. Immersive VR, typically experienced through a head-mounted display, creates a strong sense of spatial presence by tracking the user’s movements and allowing real-time interaction within the virtual environment. In contrast, non-immersive systems offer limited interaction and present the user as an external observer, reducing the feeling of being physically present in the virtual space [16]. Moreover, VR-based rehabilitation games featuring simple, goal-oriented tasks, such as drag-and-drop or point-and-click games, have been associated with improved clinical outcomes [17].

### 1.2. Virtual Reality in Motor Rehabilitation for RTT

Emerging studies have begun exploring the application of VR in motor rehabilitation for RTT patients. Stasolla et al. [18,19,20] used sensor-based systems to encourage behavioral learning in RTT, allowing patients to initiate activities such as playing videos or music through specific movements, reinforced by positive feedback. These studies demonstrated increases in accurate responses, suggesting that RTT patients can engage with and benefit from technology-driven interventions.

Mraz et al. [21] extended this work by using Microsoft Kinect^®^ and FAAST software to facilitate interaction with virtual games through body movements, without using wearable sensors. Over a 12-week intervention, RTT patients displayed improved upper limb functionality and reductions in stereotypical hand movements. Fabio et al. [22] further highlighted the motivational and emotional benefits of VR, showing that RTT patients were more engaged and responsive within 3D virtual environments compared to 2D or real-world conditions. However, this study also noted the limitations of current VR environments, which often lack multisensory stimulation and dynamic design elements essential for sustained engagement.

### 1.3. Aim

The potential of VR to address RTT’s complex rehabilitation needs lies in its adaptability, interactivity, and ability to foster cause-and-effect understanding, a critical component for therapeutic success in this population. However, further research is needed to develop RTT-specific VR applications, incorporating multisensory components and personalized task designs. For these reasons, the present study has the following aims:Evaluate the impact of a VR training program on upper limb motor skills improvement in patients with RTT, focusing on exercises involving reaching and hand-opening movements. Significant differences are expected between the experimental and control groups following the training in motor parameters, including on Temudo’s analysis of stereotypy intensity and frequency [23], Time to Satisfy Request for the first reaching task and the hand-opening task, and Correct Performances in both tasks. More specifically, we expect a significant empowering trend across all ten phases of training. Within the training program, we expect that the parameters Temudo’s Analysis of Intensity, Temudo’s Analysis of Frequency, Time to Satisfy Request, and Correct Performances will show measurable improvements in motor performance and stereotypy reduction.Assess the transferability of the results obtained from the VR training program to real-world (ecological) contexts, with a focus on upper limb motor skills. No significant differences are expected between the virtual and ecological conditions for both tasks (reaching and open hand) and for parameters including Time to Satisfy Request, Length of Reaching Movement, and Correct Performances.Evaluate the role of VR in enhancing motivation and engagement during the execution of motor tasks, addressing the main challenges of the rehabilitation process for this population. Specifically, it is anticipated that the experimental group will demonstrate improvements in attention and motivation throughout the training sessions.

Based on previous research, this study aims to provide new insights into the application of VR in RTT rehabilitation by addressing several gaps in the literature. First, it focuses on the systematic training of two specific motor functions—reaching and hand opening—within a structured VR environment. Second, it adopts a progressive intervention model based on a three-phase structure (from cause–effect interaction to functional tasks), which has not been previously applied in RTT studies. Third, the study incorporates both behavioral measures of motivation and an ecological generalization test to evaluate real-world skill transfer. Finally, it explores the potential of VR to reduce stereotypic behaviors and improve attention, offering a multidimensional view of its rehabilitative impact. These combined elements distinguish the present work from prior research and potentially enhance the evidence base for VR interventions in RTT.

## 2. Materials and Methods

### 2.1. Participants

Twenty patients with Rett Syndrome, aged between 5 and 33 years, were recruited through the Italian Rett Syndrome Association (AIRETT). Participants were randomly assigned to two groups: the Control Group (CG; n = 10, mean age = 16 years, SD = 7) and the Experimental Group (EG; n = 10, mean age = 17 years, SD = 11).

The wide age range (5–33 years) of participants reflects the real-world variability of individuals with RTT and the inclusive recruitment strategy adopted by AIRETT, aiming to involve as many families as possible. The pros are as follows: (a) the broad applicability of the intervention across developmental stages, allowing an assessment of its effects on both foundational and more advanced adaptive skills; (b) the opportunity to support participants during different life phases, considering that younger individuals may benefit from greater neural plasticity, while older ones may experience gains in autonomy and social participation; and (c) the potential to observe the long-term benefits of the intervention, particularly in adolescents and adults. The cons are (a) the need for different levels of caregiver involvement, as younger participants may require more direct support during sessions, and (b) the heterogeneity in functional abilities, which, despite the consistently severe nature of RTT, may necessitate individualized adaptations and complicate generalization of results.

Despite this variability, all participants met the diagnostic criteria for typical RTT and were evaluated to be functionally capable of engaging with the VR tasks proposed in the study. RTT patients were classified according to the classical RTT criteria outlined by Hagberg [3] as clinical stage III (characterized by prominent hand apraxia/dyspraxia, apparently preserved walking abilities, and some communicative skills, mainly through eye contact) or stage IV (late motor deterioration with progressive loss of walking ability). All participants displayed pervasive upper limb stereotypies. All participants except one (age 33) attended schools or socio-educational centers, depending all their age. The AIRETT team conducted a general assessment of the sample using the Global Assessment and Intervention in Rett Syndrome (GAIRS) Checklist [24], the Rett Assessment Rating Scale (RARS) [25], and the Downs’ Scale [26]. Table 1 and Table 2 show the characteristics of the two groups and their statistical clinical scale equivalence.

The MeCP2 mutation was identified in 100% of the sample. Patients with FOXG1 and CDKL5 mutations were excluded from the sample.

To be included in the study, participants needed to be within the age range of 3 to 35 years. They also needed to demonstrate sufficient postural control to sit independently and exhibit motor intentionality, as well as behavioral capabilities such as maintaining eye contact, engaging in joint attention, and tracking visually.

In addition, participants were selected based on high frequency and intensity of stereotypies, particularly behaviors like “hands together” or “hand-clapping on the midline”. Exclusion criteria included severe structural retractions or deformities that restricted upper limb movement, as well as drug-resistant epilepsy. Finally, participation was contingent on the availability of trained teachers who could provide ongoing support within the school setting.

Participants’ caregivers were contacted to provide a detailed explanation of the study, its objectives, procedures, and potential benefits and risks. Before starting the activities, caregivers were asked to sign an informed consent form. This document confirmed their willingness to participate in the study and certified that they fully understood the details, participation methods, and participants’ rights.

For the selected participants, during the evaluation phase before the start of the training for both groups, AIRETT therapists recorded the following demographic and clinical variables obtained through as initial assessment: age, gender, type of mutation, syndrome severity based on the Rett Assessment Rating Scale (RARS), functional level based on the Global Assessment and Intervention in Rett Syndrome (GAIRS) Scale, and manual functional motor skills level based on the Downs’ protocol [27].

The Rett Assessment Rating Scale (RARS) [25] is a standardized scale used to evaluate the severity of the disease in patients with RTT. The total score allows the measurement of the severity of the disease along a continuum ranging from mild to severe symptoms. It provides a comprehensive assessment of core symptoms across various domains, including motor skills.

The Downs’ Scale [27] for the level of purposeful hand function [26] defines the level of motricity of the hands of patients with RTT by assigning a score from 1, the minimum of manual functionality, to 8, the maximum of manual functionality.

The Global Assessment and Intervention in Rett Syndrome (GAIRS) Scale is a checklist that provides an overview of the different aspects of the syndrome and is intended for use in the functional analysis of the overall abilities of patients with RTT. The GAIRS checklist is composed of 10 macro-areas. For each area, different sequential skills, hierarchically structured, are evaluated. Each skill has a numerical score ranging from 1 to 5, where 1 is the minimum level of capacity and 5 is the maximum level of capacity to perform a specific activity.

### 2.2. Instruments

Pre-Post Test Measures (T1-T2). The measurement parameters for the test phases were collected in two distinct settings: VR and ecological (real-life) environments. The experimental test phase was conducted with the VR system, while the ecological test phase took place in a real-world setting, replicating the movements proposed in VR.

Stereotypy analysis, based on Temudo’s checklist, was performed throughout the entire test phases, repeated at both T1 and T2. Additional measures focused on two specific tasks required during the test phases: reaching movements and hand-opening movements. Hand-opening movements are characterized by the separation of the two hands, in contrast to the stereotypical motion of hand-wringing. These measures, detailed in Table 3, include the Time to Satisfy the Request for both tasks [26,27] and the Correct Performances recorded for each task.

### 2.3. Training Measures

The training sessions refer only to the experimental group. The parameters monitored for each session (4 sessions per week) were the following: Time to Satisfy Request for reaching movements and hand-opening movements and the number of Correct Performances for reaching movements and hand-opening movements, both automatically recorded by the system.

In addition to these parameters, therapists recorded the frequency and intensity of stereotypies, following Temudo’s model [23].

Behavioral aspects were also monitored by the therapists during training [28,29]. Specifically, the therapists, for the entire training session, recorded the Attention time, the Number of Aids Given and the Happiness Index, according to the scales of Van der Maat [30] and Ventura, Brivio, Riva, and Baños [16]. Table 4 describes the parameters recorded during the training sessions for the experimental group.

### 2.4. Equipment and Materials

A non-immersive VR application called “Virtual Room 2.0” was developed purposely for the training and assessment of the VR group as an extended and improved version of the one described in [22]. It was developed using the Unity 3D game engine and featured different exercises, grouped into three phases of increasing difficulty, meant to accustom the participants to the VR environment. The software was executed on a desktop or laptop computer operated by a caregiver designated to manage the session.

All the exercises exploited the same input modality based on data from a Zed Mini camera (Stereolabs Inc., San Francisco, CA, USA). The device is equipped with a pair of calibrated cameras and is therefore able to capture both images and depth data from the environment in front of it. This allowed the application of machine learning-based pose estimation algorithms to track the participants’ movements in real-time. Their inferred pose was then used to animate a human avatar, perceived in a first-person perspective in the virtual environment.

In the intermediate exercises, the input system also included two conductive pads, one hard and one soft, positioned in front of the participant. They were connected to the PC through a Click4All device, developed by Fondazione Asphi (Bologna, Italy; https://asphi.it/click4all/, accessed on 16 April 2025) and based on a concept originally developed at the Massachusetts Institute of Technology (MIT, Cambridge, MA, USA). This is a convenient interface able to use conductive objects to create keyboard-like input for a computer. The pads were presented one at a time according to VR scenario requirements.

The game was shown to the participants through a large display sized between 50″ and 65″ positioned in front of them and centered at eye level, while they were sitting on a chair about 1.5 m from the screen, with a table to rest their arms and hosting the conductive pads. The Zed Mini was placed in between, not interfering with the view of the screen, so that the participant could be correctly framed by the cameras. An example of the VR environment can be seen in Figure 1.

### 2.5. Procedure

The following figure (Figure 2) summarizes the phases of the project, and the corresponding measurements.

As seen in Figure 2, the process included the following:Sample Selection and Initial AssessmentPre-test Phase (T1)TrainingPost-test Phase (T2)

After a comprehensive assessment of participants’ functioning, in the pre-test and post-test phases both the experimental and control groups underwent the test phase, which was then repeated at the end of the training. The VR-based test session involved asking the participants to perform two hand-opening movements and four touch movements (two movements with two different targets). The ecological test session was conducted by asking the children to perform two hand-opening movements and four reaching movements. These movements corresponded to real-life actions, such as opening arms to move curtains and approaching and touching a musical instrument or an animal (realistic or a puppet). For each exercise, the therapist performed a demonstrative trial, fully assisting the participants to complete the task and reinforcing their understanding of the request. Each task was scored as follows: 1 if the child performed it independently, 0.5 if performed with partial assistance, and 0 if the goal was not achieved. The total score (maximum 6) was obtained by summing the performance of the two movements.

Before starting the training phase, brief training in the use of VR was administered to caregivers. The training required for professionals and caregivers to implement the VR protocol was minimal. All staff involved in the intervention were provided with a brief training session (approximately 2 h), which included hands-on practice with the VR system, explanation of the structured protocol, and procedures for data collection and behavioral scoring. No advanced technical skills were required. The training was conducted in-person by a member of the research team and could be replicated in other settings at minimal cost, particularly when using commercial VR devices with user-friendly interfaces. An instructional manual was also provided to support implementation fidelity.

The training phase was conducted in an everyday life setting and lasted for 2 months, focusing on practicing two target skills: reaching and hand-opening tasks. Five sessions per week were conducted at school. Based on the literature on VR intervention duration, it was determined that 720 min of treatment should be sufficient for experimental VR training [31,32]. The patients were scheduled for 40 sessions of 20 min each over an 8-week period. Given the wide variability in the clinical profiles and ages of patients with RTT, session durations were individualized for each participant based on their average attention span. The training included three phases, each designed to progressively build participants’ skills, transitioning from cause–effect interactions to targeted motor tasks. The overall goal was to promote skill mastery through VR-assisted activities tailored to the needs of each participant. Each exercise was initially conducted with total physical guidance and verbal instruction provided by the caregiver. After this first phase, the exercise was repeated three times more using only verbal instructions. Every 10 s the verbal prompt was repeated and after 120 s, if no response emerged from the participant, the exercise session was considered concluded. The verbal prompt was configured by the software system. During the training, strategies based on applied behavior analysis (ABA) were used [21,33,34,35].

For the individual using virtual reality (i.e., the participant), the primary physical challenge consisted of maintaining head and trunk control, even passively, to enable the motion tracking software to accurately identify and track body segments and joints. The motor demands imposed by the virtual reality system described in the present study are primarily related to fine motor control, as the task requires goal-directed reaching and the inhibition of stereotyped movements. For the individual administering the VR intervention (i.e., the caregiver), there were no significant physical challenges associated with the setup or operation of the system.

Among the medications commonly used in individuals with Rett Syndrome that could potentially influence the outcome are antiepileptic drugs. If inappropriately dosed, these can induce somnolence and consequently reduce the individual’s ability to actively participate in the intervention. Caregivers were instructed to administer the intervention sessions during the participants’ alert and wakeful periods.

During the training phase, participants were required not to wear upper limb positioning splints, which are typically used to limit stereotypic movements. By removing the splints, we ensured that any reduction in stereotypies could not be attributed to mechanical constraints imposed by the orthoses.

### 2.6. Statistical Analysis

Non-parametric tests were used for statistical analysis as the data did not meet parametric assumptions. Although the CG and EG were equivalent at baseline, non-parametric methods were applied. Within-group comparisons used the Wilcoxon signed-rank test, between-group comparisons employed the Mann–Whitney U test, and trends over 10 training sessions were assessed with the Friedman test.

## 3. Results

Regarding the first hypothesis, the VR training program significantly reduced the frequency and intensity of stereotyped movements in the EG, with no changes in the CG. Wilcoxon signed-rank tests confirmed improvements within the EG (*p* < 0.01 for frequency, *p* = 0.03 for intensity) and significant between-group differences, with the EG outperforming the CG for the hand-opening and reaching tasks (*p* < 0.01 for frequency, *p* = 0.03 for intensity).

The EG also showed a significant post-training reduction in response time for both tasks, while no changes occurred in the CG (*p* < 0.01, Wilcoxon; *p* < 0.01, Mann–Whitney U). Additionally, the number of correct responses significantly increased in the EG, with no improvements in the CG (*p* < 0.01, Wilcoxon; *p* < 0.01, Mann–Whitney U) (Table 5).

Regarding the second part of the first hypothesis, the Friedman test revealed significant trends across the 10 training sessions for multiple parameters, indicating consistent improvements. Significant results were observed for Temudo’s Analysis of Intensity (χ^2^(9) = 35.76, *p* = 0.001), Temudo’s Frequency Analysis (χ^2^(9) = 30.54, *p* = 0.002), Time to Satisfy Request (χ^2^(9) = 28.83, *p* = 0.003) and Correct Performances (χ^2^(9) = 33.15, *p* = 0.001). These findings support the efficacy of the VR training program in enhancing motor skills and reducing stereotypies, with progressive improvements observed throughout the sessions (Table 6).

The second hypothesis examined the generalization of VR training results to real-world (ecological) contexts, focusing on motor skills in reaching tasks. The EG showed a significant reduction in response time (Time to Satisfy Request) from pre- to post-test in both the virtual and ecological conditions. Wilcoxon tests confirmed significant improvements within the EG (*p* < 0.01), while Mann–Whitney U tests highlighted significant differences between the groups (*p* < 0.01).

However, the differences between the virtual and ecological conditions were not statistically significant in the post-test phase (*p* = 0.84, Mann–Whitney). Significant improvements were observed in Correct Performance, with the experimental group showing higher effectiveness in virtual conditions (*p* < 0.01, Wilcoxon; *p* < 0.01, Mann–Whitney U). These improvements were specific to the virtual context and were not replicated in the ecological setting (Table 7).

The third hypothesis examined VR’s role in enhancing motivation and engagement during motor tasks in RTT rehabilitation. Trend analysis showed significant results, with the EG demonstrating increased attention and motivation throughout the sessions, along with a reduction in the number of aids required. Table 8 presents the medians and range for each phase.

Trend analysis for the Attention Time parameter shows a clear increase over the training sessions, with attention duration rising from 4 s in the first session to 300 s in the last. Non-parametric trend analysis (e.g., the Friedman test or Cochran’s Q test) confirmed a significant increase in attention over time (*p* < 0.01), indicating greater engagement and sustained focus by participants.

The Number of Aids Given showed a clear downward trend, with a reduction from a median of 32 (range 12–64) in the first session to 18 (range 7–48) in the tenth. Trend analysis (e.g., the Friedman test) confirmed a significant decrease in the need for aids (*p* < 05), indicating increasing independence as motor and cognitive engagement improved.

Regarding the Motivation Index, motivation remained stable, with median values around 5.5 to 6 (range 4–7) throughout the sessions. Trend analysis revealed no significant change in motivation (*p* > 0.05), suggesting consistent engagement throughout the training. Throughout the VR training sessions, caregivers (usually parents) were consistently present and provided emotional support, verbal encouragement, and non-verbal reassurance. Although their role was not formally assessed, their involvement appeared to support children’s motivation and sustained engagement during the tasks. This presence might have contributed to the children’s willingness to participate and complete the sessions, especially in moments of initial hesitation or fatigue. The social and emotional scaffolding provided by caregivers should be considered an important contextual factor in interpreting the results, particularly in relation to motivation and behavioral engagement.

In conclusion, the results support the hypothesis: attention and motivation remained stable or slightly improved, while the number of aids required significantly decreased, reflecting a positive impact of the VR training on task execution and reduced reliance on external support. The trend analysis confirmed significant findings in attention (*p* < 0.01) and aids (*p* < 0.05), while motivation remained stable (*p* > 0.05). Further analysis is needed to explore factors contributing to these trends, particularly in relation to the VR program’s design and individual responses.

## 4. Discussion

This study aimed to evaluate the effectiveness of a VR training program in enhancing upper limb motor skills, promoting the generalization of these improvements to real-world contexts, and fostering motivation and engagement during rehabilitation in individuals with RTT. The findings support the VR intervention’s efficacy, showing improvements in motor skills, reduction in stereotypical behaviors, and enhanced rehabilitation experience, in line with previous research on VR in RTT and other neurodevelopmental disorders.

For the first hypothesis, the results showed significant improvements in the frequency and intensity of stereotypical hand movements in the experimental group, aligning with the findings of Stasolla et al. [18,19,20] and Mraz et al. [21], who found that VR effectively reduces stereotypical behaviors and improves motor functions in RTT. The control group showed no notable changes, reinforcing the VR intervention’s impact as opposed to traditional therapies.

Trend analysis using the Friedman test indicated significant improvements across all measured motor parameters, including Temudo’s analysis of intensity and frequency, Time to Satisfy Requests and Correct Performance. These results align with Fabio et al. [22], who highlighted the motivational benefits of immersive VR environments. The observed reduction in stereotypic behaviors among participants may be explained by several interrelated mechanisms. First, VR provides a highly engaging and immersive environment that captures attention and promotes sustained focus on goal-directed tasks. This increase in attentional engagement may temporarily inhibit the automatic emergence of stereotypies, which often arise during low-demand or passive moments. Second, the structured and rewarding nature of VR activities, especially those that include cause–effect interactions and clear feedback, may enhance cognitive and motor planning, thereby fostering better functional motor patterns. Third, by offering immediate and multisensory feedback (visual, auditory, and sometimes tactile), VR can reinforce purposeful movements, which may gradually compete with or replace maladaptive repetitive behaviors. Importantly, during the initial phases of training, each task was supported by a caregiver who provided full physical and verbal assistance. As participants became more familiar with the tasks, the level of support was gradually reduced, promoting increased autonomy and reinforcing self-initiated, functional actions. Finally, the use of ABA-based reinforcement strategies embedded in the VR protocol further contributed to reducing stereotypies, as these behaviors were not reinforced, while goal-directed actions were consistently encouraged and rewarded. These combined factors may explain the progressive decline in stereotypic behavior observed in the experimental group.

Regarding the second hypothesis, the results for the transferability of VR training to real-world contexts were mixed. While significant improvements were observed in virtual conditions (e.g., Time to Satisfy Request and hand-opening), these gains did not always generalize to ecological conditions, aligning with the challenges noted by Franze et al. [15]. Nevertheless, the observed improvements suggest VR’s potential to enhance functional abilities in various settings, a crucial factor for successful RTT rehabilitation, particularly for individuals in remote areas [8]. Moreover, although the present study focused specifically on upper limb motor functions, attention, and stereotypies, qualitative observations made by therapists and caregivers occasionally suggested subtle improvements in participants’ overall engagement and intentional communication. While no standardized assessments of gross motor or communicative abilities were conducted, these anecdotal impressions may point to broader benefits of the VR training and support the need for future studies to include these domains in a more systematic way.

Overall, participants were able to engage with the VR activities with varying degrees of support. The system’s non-immersive nature (large screen and first-person perspective) minimized sensory overload and was generally well tolerated, even by participants with higher stereotypy levels. Moreover, a qualitative age-related difference in response to the VR intervention was observed, although this difference was not statistically significant. Younger participants appeared to engage more readily with the virtual tasks, likely due to their greater familiarity with digital technologies from an early age (e.g., tablets, touchscreens, and interactive applications). In contrast, older participants, especially those with limited prior exposure to such tools, showed initial difficulties and a slower start in the activities. This suggests that technological familiarity, rather than age per se, may play a key role in how effectively individuals with RTT interact with VR-based interventions. These findings, though not quantitatively confirmed, highlight the importance of tailoring the introduction and pacing of technology-based therapies according to the participant’s previous experiences with similar devices.

While the findings of this study are promising, several limitations must be acknowledged. First, the small sample size of 20 participants restricts the generalizability of the results to the broader population of individuals with RTT. Furthermore, the wide age range of participants (5–33 years) may have introduced variability in developmental stages, including the onset of puberty and its associated cognitive and behavioral changes. Although an exploratory analysis comparing younger (5–13 years) and adolescent (13–19 years) participants did not reveal statistically significant differences in outcomes, a slight trend toward greater improvements was observed in the younger subgroup. Due to the limited sample size, these findings must be interpreted with caution. Future studies with larger, age-stratified samples are needed to better understand the influence of developmental stage on VR-based rehabilitation outcomes in individuals with RTT. Third, the intervention duration of 40 sessions over eight weeks, while sufficient to observe initial improvements, may not provide a comprehensive understanding of the long-term impact of VR training. A longer intervention period might clarify whether these benefits can be sustained or amplified over time. Additionally, the study lacked a long-term follow-up phase. The absence of evaluations beyond the immediate post-training period makes it challenging to assess the durability of the observed improvements or to determine the extent to which skills learned in VR translate to real-world contexts over time. Incorporating follow-up assessments three months or more post intervention would provide valuable insights into the lasting effects of VR rehabilitation. Moreover, the pre- and post-test sessions were specifically designed for this study to evaluate functional improvements in reaching and hand-opening tasks. While these tasks were inspired by real-life actions, to ensure ecological validity they were not derived from standardized clinical tests. However, the scoring method (independent/partial assistance/failure) was based on commonly used behavioral observation techniques in motor rehabilitation and early intervention studies. Inter-rater reliability was ensured by having the same trained therapist conduct and score all test sessions. Future studies should consider integrating standardized motor assessments to further validate these outcome measures.

Motivation also presented a limitation. The stability of the motivation index suggests a potential ceiling effect, as participants were already highly motivated at the start of the program. This limited the ability to assess whether VR could enhance motivation further, despite its engaging nature. Finally, the VR environment itself could be improved by incorporating more multisensory stimuli and dynamic design elements. These features could increase engagement, sustain motivation over longer periods, and make the VR experience more immersive, ultimately enhancing the rehabilitation process.

Addressing these limitations in future research will be critical to maximizing the effectiveness of VR interventions for individuals with RTT and ensuring their applicability to real-world rehabilitation scenarios.

## 5. Conclusions

The findings suggest that VR training can effectively enhance upper limb motor skills, reduce stereotypical behaviors, and promote engagement in rehabilitation for individuals with RTT. These results align with previous findings on VR’s potential in RTT rehabilitation, such as in studies by Stasolla et al. [18], Fabio et al. [22], and Mraz et al. [21]. Significant improvements in motor performance and a reduction in the number of aids required highlight VR’s potential to address core challenges in RTT rehabilitation, as noted by Lotan and Ben-Zeev [5]. Although the generalization of improvements to real-world contexts was not always observed, the findings support integrating VR into rehabilitation programs for RTT, as advocated by Caprì et al. [8].

Future research should focus on refining VR environments to better support skill transfer to real-world situations, incorporating multisensory components and individualized task designs to enhance motor outcomes and engagement. Additionally, studies should explore strategies to sustain and improve motivation over extended rehabilitation periods, ensuring that VR interventions provide long-term benefits. As Kinnunen et al. [10] emphasized, such research is crucial for improving developmental outcomes and offering innovative solutions for RTT rehabilitation.

## Figures and Tables

**Figure 1 pediatrrep-17-00049-f001:**
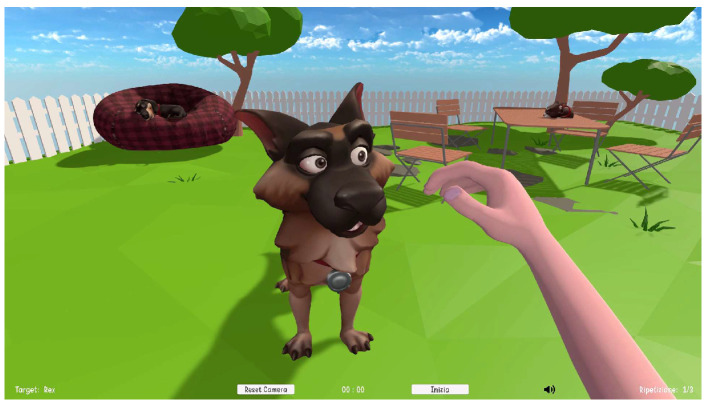
Example of the VR environment, where the requested task is to pet a virtual dog.

**Figure 2 pediatrrep-17-00049-f002:**
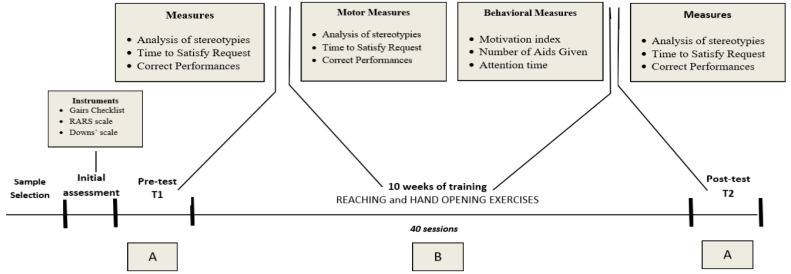
Overview of the Experimental Protocol and Measurements. The figure illustrates the structure of the study, including the sample selection, initial assessment, pre-test (T1), 10-week training period, and post-test (T2). The training consisted of 40 sessions of exercises and contrasting stereotypies.

**Table 1 pediatrrep-17-00049-t001:** Characteristics of participants.

Patient ID	Name	Clinical Stage	Age (Years)	MeCP2 Mutation	Level of Severity (RARS)	Functional Ability Level (GAIRS)	Downs’ Scale
EG							
1	L.M.	III	10	C965C	58.0	1.85	1
2	L.A.	IV	27	T158M	69.5	2.45	1
3	L.G	IV	27	T158M	69.5	2.26	1
4	V.S.	II	5	C538C	67.5	1.80	2
5	D.D.	IV	33	R306C	67.5	2.48	2
6	B.C.	II	8	C965C	89.5	2.07	3
7	B.A.	III	16	P152R	57.0	2.10	1
8	A.G.	IV	28	R270X	67.0	2.43	1
9	H.G.	II	7	T158M	71.0	1.84	1
10	B.M.	II	7	C502C	79.0	2.10	1
CG							
1	C.A.	II	8	C502C	58.0	2.55	2
2	C.M.	III	10	T158M	65.5	1.90	1
3	A.A.	III	13	C763C	85.5	1.50	1
4	A.S.	IV	18	C1156	57.0	2.34	2
5	G.B.	IV	27	P152R	63.5	2.00	3
6	L.G.	III	15	R106W	64.0	2.70	3
7	L.S.	III	12	T158C	67.5	1.40	2
8	S.D.	IV	18	P133C	72.0	2.19	4
9	D.F.	IV	28	R255X	64.5	1.69	2
10	B.E.	II	9	C880C	58.0	2.15	1

EG: experimental group; CG: control group.

**Table 2 pediatrrep-17-00049-t002:** Clinical descriptions of the experimental group (EG) and the control group (CG). Data are reported as mean ± standard deviation).

Participants	EG N = 10	CG N = 10	*p*-Value *
Age (Years)	17 ± 11	16 ± 7	
Gairs Checklist	2 ± 0.26	2 ± 0.43	0.55
RARS Scale	66 ± 6	65 ± 8	0.09
Downs’ Scale	1.4 ± 0.7	2.1 ± 1	0.09
Right Lateral Dominance	9	9	
Left Lateral Dominance	1	1	

* *p*-value are based on Mann–Whitney U test.

**Table 3 pediatrrep-17-00049-t003:** Pre-post test measures.

Measure	Description	Data Recording
Temudo’s Checklist	It is used to assess the presence, frequency, and characteristics of motor stereotypies, distinctive symptoms of RTT. It categorizes stereotypies based on the nature of movements, such as hand-wringing, hand-clapping, bringing hands to the mouth, and other repetitive behaviors [23]	Recorded by the therapist.
Time to Satisfy Request	It assesses the patient’s reaction times to stimuli.	Automatically recorded by the software.
Correct Performances	Number of correct responses performed by the participant during a series of trials.	Automatically recorded by the software.

**Table 4 pediatrrep-17-00049-t004:** Pre-post test measures.

Measure	Description	Data Recording
Analysis of stereotypies according to Temudo’s checklist	Described in Table 3	Recorded by the therapist.
Time to Satisfy Request for reaching movements and hand-opening movements	Described in Table 3	Automatically recorded by the software.
Correct Performances for reaching movements and hand-opening movements	Described in Table 3	Automatically recorded by the software.
Number of Aids Given for all training sessions	It refers to the frequency of cues provided to participants during all training sessions.	Recorded by the therapist.
Attention time for all training sessions	It refers to the total duration participants maintained focus during all training sessions.	Recorded by the therapist.
Motivation index (MI) for all training sessions	It is based on Van der Maat’s [30] taxonomy and assesses motivation through five behavioral categories: gaze direction, sounds, mouth movements, physiological reactions (e.g., blushing, sweating), and hand gestures. The motivation index was calculated by summing the scores for these five behaviors. These behaviors were recorded via a front-facing camera. Two independent observers evaluated the videos, with a concordance index of at least 95.	Recorded by the therapist.

**Table 5 pediatrrep-17-00049-t005:** Median (range min–max) and *p*-values (Mann–Whitney U test) for EG and CG in pre-test and post-test phases.

Parameter	Group	Pre-Test	Post-Test	*p* (Pre vs. Post)
Temudo’s Analysis of Stereotypy Intensity	EG	24 (10–48)	15 (5–30)	0.008 **
	CG	25 (12–60)	24 (10–55)	0.350
Temudo’s Analysis of Stereotypy Frequency	EG	210 (120–350)	145 (70–240)	0.005 **
	CG	215 (90–390)	214 (100–380)	0.240
Time to Satisfy Request for Reaching Task	EG	86 (40–140)	56 (30–98)	0.012 *
	CG	84 (45–130)	78 (40–110)	0.870
Time to Satisfy Request for Hand-Opening	EG	113 (90–140)	98 (85–125)	0.030 *
	CG	134 (90–140)	116 (80–125)	0.780
Correct Performances (Reaching Task)	EG	0.40 (0.10–0.80)	0.80 (0.40–1.20)	0.003 **
	CG	0.45 (0.20–0.90)	0.46 (0.25–0.90)	0.690
Correct Performances (Hand-Opening Task)	EG	0.42 (0.15–0.70)	0.82 (0.60–1.00)	0.002 **
	CG	0.44 (0.10–0.85)	0.46 (0.20–0.80)	0.620

Note: * = *p* < 0.05; ** = *p* < 0.01.

**Table 6 pediatrrep-17-00049-t006:** Comparison of medians and ranges for the experimental group during training sessions.

Parameter	Session 1	Session 2	Session 3	Session 4	Session 5	Session 6	Session 7	Session 8	Session 9	Session 10
Reaching Task—Motor Aspects										
Temudo’s Intensity	120 (35–260)	125 (50–300)	130 (40–250)	120 (45–250)	130 (50–300)	120 (45–250)	115 (50–240)	115 (50–220)	105 (50–200)	80 (30–180)
Temudo’s Frequency	13 (5–30)	14 (5–40)	13 (5–35)	12 (5–30)	12 (5–30)	11 (5–30)	11 (5–30)	11 (5–30)	10 (5–20)	7 (4–15)
Time to Satisfy Request	55 (30–100)	50 (30–100	45 (30–90)	40 (30–80)	40 (20–80)	35 (20–80)	33 (20–70)	32 (20–70)	30 (20–60)	25 (15–50)
Correct Performances	5 (3–9)	6 (3–9)	7 (4–9)	7 (4–9)	7 (4–9)	7 (4–9)	7 (4–9)	7 (4–9)	8 (5–9)	8 (5–9)
Hand-Opening Task—Motor Aspects										
Temudo’s Intensity	240 (90–400)	240 (100–400)	200 (90–400)	190 (90–400)	170 (70–400)	160 (70–350)	140 (50–250)	120 (50–250)	100 (50–250)	100 (50–250)
Temudo’s Frequency	15 (7–30)	16 (7–40)	14 (7–30)	14 (7–30)	14 (7–30)	12 (6–25)	11 (5–20)	9 (5–15)	9 (5–15)	8 (5–10)
Time to Satisfy Request	55 (30–110)	50 (30–100)	40 (20–100)	40 (20–90)	40 (20–100)	35 (20–90)	30 (20–80)	30 (20–80)	30 (20–70)	25 (15–50)
Correct Performances	1 (0–3)	2 (0–3)	3 (1–5)	4 (2–6)	4 (2–6)	4 (2–6)	4 (2–6)	5 (3–6)	5 (3–6)	5 (3–6)

**Table 7 pediatrrep-17-00049-t007:** Median (range) and *p*-values (Wilcoxon test) for the experimental group in virtual and ecological conditions.

Parameters	Phases	Virtual	Ecological	*p* (Virtual vs. Ecological)
Temudo’s Analysis of Stereotypy Intensity	Pre-Test	24 (12–46)	28 (20–48)	0.210
	Post-Test	15 (9–30)	16 (10–42)	0.840
Temudo’s Analysis of Stereotypy Frequency	Pre-Test	210 (120–320)	220 (120–350)	0.130
	Post-Test	145 (100–220)	149 (110–250)	0.790
Time to Satisfy Request for Reaching Task	Pre-Test	86 (38–140)	92 (40–140)	0.120
	Post-Test	56 (30–110)	55 (42–130)	0770
Time to Satisfy Request for Hands-Opening	Pre-Test	113 (88–142)	120 (90–170)	0.140
	Post-Test	98 (80–128)	100 (92–135)	0.560
Correct Performances (Reaching Task)	Pre-Test	0.40 (0.12–0.78)	0.40 (0.10–0.80)	0.910
	Post-Test	0.80 (0.55–0.95)	0.72 (0.20–0.85)	0.130
Correct Performances (Hands-Opening Task)	Pre-Test	0.42 (0.10–0.72)	0.42 (0.15–0.70)	0.980
	Post-Test	0.82 (0.50–0.85)	0.72 (0.20–0.72)	0.180

**Table 8 pediatrrep-17-00049-t008:** Medians and ranges (min–max) of Attention Time, Number of Aids Given During Task Performance, and Motivation Index across training sessions.

Parameter	Session 1	Session 2	Session 3	Session 4	Session 5	Session 6	Session 7	Session 8	Session 9	Session 10
Attention Time (seconds)	4 (4–6)	10 (6–12)	20 (12–30)	40 (30–50)	60 (50–70)	90 (70–110)	120 (100–140)	150 (130–170)	200 (180–220)	300 (250–300)
Number of Aids Given	32 (12–64)	35 (14–70)	32 (16–68)	31 (13–64)	25 (10–60)	25 (11–60)	22 (10–60)	20 (8–56)	20 (8–56)	18 (7–48)
Motivation Index	5.5 (4–7)	5.5 (4–7)	6.0 (5–7)	5.5 (4–7)	6.0 (5–7)	6.0 (5–7)	6.0 (5–7)	5.5 (4–7)	6.0 (5–7)	6.0 (5–7)

## Data Availability

Data are available as Appendix A of the present article.

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
