# Peer review of "Virtual Reality as a Tool for Upper Limb Rehabilitation in Rett Syndrome: Reducing Stereotypies and Improving Motor Skills"

_pediatrrep, 2025, doi:10.3390/pediatric17020049_

Round 1

Reviewer 1 Report

Comments and Suggestions for Authors

Wide variation in age of subjects: 5-33. Could this be further analyzed to reflect changes between 5-13 years and 13-19 years with impact on puberty.

Further details about parenting or caregiver involvement in motivation. Does this affect outcome?

Comment on any changes in Gross motor and communication.

Further comment on Pre and post tests to assess validity of the tests. What training is required for PT and Psychology and what is the cost of the training?

Motivation levels remained stable.-explain and equate to impact on the study.

Are there any physical challenges in using VR?

More details how VR reduces stereotypic behavior in Rett. 

Is there any drop off after VR training after 8 weeks?

Did any medication used by Rett influence outcome? Any splints used influence outcome?

Comments on the Quality of English Language

Minor changes.

Author Response

Comment 1

Wide variation in age of subjects: 5-33. Could this be further analyzed to reflect changes between 5-13 years and 13-19 years with impact on puberty.

Response1

Thank you for your valuable observation regarding the wide age range of participants (5–33 years) and the potential impact of developmental stages, particularly puberty. As suggested, we conducted an exploratory analysis by dividing participants of the experimental group into two subgroups: 5–13 years and 13–19 years. The results did not reveal statistically significant differences between the two age groups in terms of treatment outcomes. However, a slight trend toward greater improvement was observed among the younger participants (5–13 years), particularly in attention time and reduction of stereotypies. Given the limited sample size and the exploratory nature of this analysis, we recognize that age-related factors, may play a role and represent a limitation of the current study. This has been acknowledged and discussed in the Limitations section of the revised manuscript (highlighted it in yellow).

Comment 2

Further details about parenting or caregiver involvement in motivation. Does this affect outcome?

Response2

We thank the reviewer for this insightful comment. In our study, caregivers were present during the VR training sessions and provided emotional support and encouragement, which we believe contributed to maintaining the children's motivation and engagement. While caregiver involvement was not systematically manipulated or measured, our clinical observations suggest that their presence had a positive impact on the participants’ persistence during tasks. We have now added this information to the manuscript in the result section (highlighted in yellow).

Comment 3

Comment on any changes in Gross motor and communication.

Response 3

Thank you for your insightful question regarding potential changes in gross motor and communication abilities. While the primary focus of this study was on upper limb function, attention, and stereotypies, we also observed participants' general behaviors during sessions. No formal assessment tools for gross motor or communication skills were included in the study protocol. However, qualitative observations from therapists and caregivers did not indicate significant changes in gross motor functioning. With regard to communication, although not systematically measured, several caregivers reported increased attempts at intentional communication and interaction, possibly as a secondary effect of the structured and engaging VR sessions. These anecdotal findings have been mentioned in the revised Discussion section (highlighted in yellow) to provide a broader perspective on potential benefits and future directions.

Comment 4

Further comment on Pre and post-tests to assess validity of the tests. What training is required for PT and Psychology and what is the cost of the training?

Response 4

The pre- and post-test sessions were specifically designed for this study to evaluate functional improvements in reaching and hand-opening tasks. While these tasks were inspired by real-life actions to ensure ecological validity, they were not derived from standardized clinical tests. However, the scoring method (independent/partial assistance/failure) was based on commonly used behavioral observation techniques in motor rehabilitation and early intervention studies. Inter-rater reliability was ensured by having the same trained therapist conduct and score all test sessions. Future studies should consider integrating standardized motor assessments to further validate these outcome measures.

The training required for professionals (psychologists, physiotherapists, and educators) to implement the VR protocol is minimal. All staff involved in the intervention were provided with a brief training session (approximately 2 hours), which included hands-on practice with the VR system, explanation of the structured protocol, and procedures for data collection and behavioral scoring. No advanced technical skills were required. The training was conducted in-person by a member of the research team and could be replicated in other settings at minimal cost, particularly when using commercial VR devices with user-friendly interfaces. An instructional manual was also provided to support implementation fidelity (we added these clarifications in the procedure section, highlighted in yellow).

With reference to the cost of the training no additional financial investment was required. The training for both physical therapists and psychologists was conducted internally and based on their existing clinical competencies in managing individuals with Rett Syndrome and in applying behavioral techniques. Familiarization with the VR system required only a short introductory session (approximately 2 hours) and ongoing supervision by the research team. Therefore, the cost was minimal and mainly involved organizational time rather than financial resources, highlighting the feasibility and sustainability of implementing VR training within standard clinical practice.

Comment 5

Motivation levels remained stable. -explain and equate to impact on the study.

Response 5

Thank you for your comment regarding the stability of motivation levels throughout the intervention. As already reported in the discussion section, participants showed high levels of motivation already at the pre-test phase, which likely led to a ceiling effect that limited the detection of further increases. This suggests that the VR intervention may not have enhanced motivation in measurable terms, but it is important to note that it was effective in maintaining high engagement across sessions, which is particularly meaningful in populations with severe motor and communicative impairments such as Rett Syndrome. Therefore, the stable motivation levels may reflect the ability of VR to sustain participant involvement, which indirectly supports its feasibility and acceptability as a rehabilitation tool. This point has now been clarified and expanded in the Discussion section.

Comment 6

Are there any physical challenges in using VR?

Response 6

Thank you. For the individual using virtual reality (i.e., the participant), the primary physical challenge consists in maintaining head and trunk control, even passively, to enable the motion tracking software to accurately identify and track body segments and joints. The motor demands imposed by the virtual reality system described in the present study are primarily related to fine motor control, as the task requires goal-directed reaching and the inhibition of stereotyped movements. For the individual administering the VR intervention (i.e., the caregiver), there are no significant physical challenges associated with the setup or operation of the system.

Comment 7

More details how VR reduces stereotypic behavior in Rett. 

Response 7

Thank you. The observed reduction in stereotypic behaviors among participants may be explained by several interrelated mechanisms. First, VR provides a highly engaging and immersive environment that captures attention and promotes sustained focus on goal-directed tasks. This increase in attentional engagement may temporarily inhibit the automatic emergence of stereotypies, which often arise during low-demand or passive moments. Second, the structured and rewarding nature of VR activities, especially those that include cause-effect interactions and clear feedback, may enhance cognitive and motor planning, thereby fostering more functional motor patterns. Third, by offering immediate and multisensory feedback (visual, auditory, and sometimes tactile), VR can reinforce purposeful movements, which may gradually compete with or replace maladaptive repetitive behaviors. Importantly, during the initial phases of training, each task was supported by a caregiver who provided full physical and verbal assistance. As participants became more familiar with the tasks, the level of support was gradually faded, promoting increased autonomy and reinforcing self-initiated, functional actions. Finally, the use of ABA-based reinforcement strategies embedded in the VR protocol further contributed to reducing stereotypies, as these behaviors were not reinforced, while goal-directed actions were consistently encouraged and rewarded. These combined factors may explain the progressive decline in stereotypic behavior observed in the experimental group.We added these observation in the discussion section, highligthed in yellow.

Comment 8

Is there any drop off after VR training after 8 weeks?

Comment 9

Did any medication used by Rett influence outcome? Any splints used influence outcome?

Response 9

Among the medications commonly used in individuals with Rett syndrome that could potentially influence the outcome are antiepileptic drugs. If inappropriately dosed, these can induce somnolence and consequently reduce the individual's ability to actively participate in the intervention. Caregivers were instructed to administer the intervention sessions during the participants' alert and wakeful periods.

During the training phase, participants were required not to wear upper limb positioning splints, which are typically used to limit stereotypic movements. By removing the splints, we ensured that any reduction in stereotypies could not be attributed to mechanical constraint imposed by the orthoses.

Reviewer 2 Report

Comments and Suggestions for Authors

The manuscript titled “Virtual Reality as a Tool for Upper Limb Rehabilitation in Rett Syndrome: Reducing Stereotypies and Improving Motor Skills” presents the development and validation of a virtual reality (VR) system specifically designed for the rehabilitation of patients with Rett syndrome. Rett syndrome is a rare genetic neurological and developmental disorder that leads to a progressive loss of motor skills and language abilities. As there is no cure for the condition, rehabilitation is a crucial aspect of care, with current treatment efforts focusing on improving movement and communication.

The paper is well-written and provides comprehensive details of the study, with conclusions that are well-supported by the presented results. The authors demonstrate that the number of aids required to perform standardized tasks significantly decreased in the VR group, suggesting a positive impact of the training on task execution and a reduction in the reliance on external support.

However, there are a few minor points that could be improved:

  1. Representation of Numbers: In Table 1, the Functional Ability Level (GAIRS) values are presented with too many significant digits. The authors may consider rounding these to just one decimal place for clarity. Similarly, in Table 2, the precision of the numbers appears excessive. It would be more appropriate to round the values so that their precision is consistent with their accuracy. For example, instead of presenting 65.55 ± 8.42, this could be simplified to 66 ± 8.

  2. In the Section 1.3, the authors state that: “Based on previous research, this study aims to provide new insights into the application of VR in RTT rehabilitation.” However, I would expect a clearer description of what differentiates this study from previous research and what specific new insights it provides. While this may be discussed later in the manuscript, it currently feels somewhat vague and could be more explicitly outlined.

  3. Line 142: Non-standard abbreviations such as EG, CG, and M are introduced without definitions. These terms should be clearly defined upon first use to ensure clarity for readers who may not be familiar with the abbreviations.

Author Response

REFEREE n. 2

The manuscript titled “Virtual Reality as a Tool for Upper Limb Rehabilitation in Rett Syndrome: Reducing Stereotypies and Improving Motor Skills” presents the development and validation of a virtual reality (VR) system specifically designed for the rehabilitation of patients with Rett syndrome. Rett syndrome is a rare genetic neurological and developmental disorder that leads to a progressive loss of motor skills and language abilities. As there is no cure for the condition, rehabilitation is a crucial aspect of care, with current treatment efforts focusing on improving movement and communication.

The paper is well-written and provides comprehensive details of the study, with conclusions that are well-supported by the presented results. The authors demonstrate that the number of aids required to perform standardized tasks significantly decreased in the VR group, suggesting a positive impact of the training on task execution and a reduction in the reliance on external support.

However, there are a few minor points that could be improved:

Comment 1

Representation of Numbers: In Table 1, the Functional Ability Level (GAIRS) values are presented with too many significant digits. The authors may consider rounding these to just one decimal place for clarity. Similarly, in Table 2, the precision of the numbers appears excessive. It would be more appropriate to round the values so that their precision is consistent with their accuracy. For example, instead of presenting 65.55 ± 8.42, this could be simplified to 66 ± 8.

Response 1

Thank you. Me modified the table

Comment 2

In the Section 1.3, the authors state that: “Based on previous research, this study aims to provide new insights into the application of VR in RTT rehabilitation.” However, I would expect a clearer description of what differentiates this study from previous research and what specific new insights it provides. While this may be discussed later in the manuscript, it currently feels somewhat vague and could be more explicitly outlined.

Response 2

Thank you for your constructive comment. We agree that a more explicit clarification of the study's novelty would strengthen the Introduction. Accordingly, we revised Section 1.3 to better highlight the distinctive features of our study in comparison to prior research. Specifically, our study provides new insights by: (1) applying a structured VR training focused on two target skills (reaching and hand-opening) in individuals with Rett Syndrome (RTT), (2) integrating a three-phase progression from cause-effect to goal-directed motor tasks, (3) incorporating a motivational index and real-world generalization assessment, and (4) observing changes in stereotypic behaviors and attention during a systematic school-based intervention. To our knowledge, no previous studies have combined these elements in a VR-based program for individuals with RTT. These aspects have now been explicitly addressed in the revised manuscript.

Comment 3

Line 142: Non-standard abbreviations such as EG, CG, and M are introduced without definitions. These terms should be clearly defined upon first use to ensure clarity for readers who may not be familiar with the abbreviations.

Response 3

Thank you for your observation. We have now defined all abbreviations (EG, CG, M, SD) upon their first appearance in the manuscript to ensure clarity for all readers.

Reviewer 3 Report

Comments and Suggestions for Authors

The authors present a manuscript describing virtual reality as a tool for upper limb rehabilitation. Overall, the manuscript will make a contribution to the field, especially about using technology in rehabilitation. Some suggestions to improve clarity:

Introduction:

-page 2, line 45: The sentence "Additional hallmark features needs citation.

-page 2, line 88: What is meant by "immersive VR"? Please provide further details and a citation.

-page 3, line 124-127: Please provide more details to describe the terms mentioned in this section about training...I.e. Temudo's Analysis of Intensity, Temundo's Analysis of frequency...etc. Are these all phases of training? Citation needed as well.

Materials and Methods:

-page 4 , line 158: Participant characteristics- you have a wide age range (5-33 years) which may mean different experiences with VR as well as time spent living with Rett Syndrome. Please explain in the methods why you have such a large age range. Also, on page 9, line 289 you say "Given the wide variability of clinical profiles.." but nothing about age.

Discussion:

-page 13, Line 403: Please mention something about the wide range of age in your study as a limitation here.  

Comments on the Quality of English Language

English language appears good but it is always helpful to have a review.

Author Response

REFEREE N. 3

The authors present a manuscript describing virtual reality as a tool for upper limb rehabilitation. Overall, the manuscript will make a contribution to the field, especially about using technology in rehabilitation. Some suggestions to improve clarity:

Comment 1

Introduction:

-page 2, line 45: The sentence "Additional hallmark features needs citation.

Response 1

Citation added.

Comment 2

-page 2, line 88: What is meant by "immersive VR"? Please provide further details and a citation.

Response 2

Thank you for the question.

Immersive and non-immersive media differ in the user’s perspective and the level of engagement they provide. Immersive VR, typically experienced through a head-mounted display, creates a strong sense of spatial presence by tracking the user's movements and allowing real-time interaction within the virtual environment. In contrast, non-immersive systems offer limited interaction and present the user as an external observer, reducing the feeling of being physically present in the virtual space. We added the citation.

Comment 3

-page 3, line 124-127: Please provide more details to describe the terms mentioned in this section about training...I.e. Temudo's Analysis of Intensity, Temundo's Analysis of frequency...etc. Are these all phases of training? Citation needed as well.

Response 3

These are the motor parameters collected during the training that will be described later in the manuscript in the Instruments part.

Citation added.

Comment 4

Materials and Methods:

-page 4 , line 158: Participant characteristics- you have a wide age range (5-33 years) which may mean different experiences with VR as well as time spent living with Rett Syndrome. Please explain in the methods why you have such a large age range. Also, on page 9, line 289 you say "Given the wide variability of clinical profiles.." but nothing about age.

Response 4

Thank you for your thoughtful observation. We have now clarified in the Methods section the rationale for including a wide age range (5–33 years), which reflects the real-world heterogeneity of individuals with RTT and the inclusive recruitment criteria of AIRETT. We also added a comment in the Discussion section to explicitly acknowledge age variability as a potential influencing factor alongside clinical diversity. They are both highlighted in yellow.

Comment 5

Discussion:

-page 13, Line 403: Please mention something about the wide range of age in your study as a limitation here.  

Response 5

We added it as limitation (highlighted in yellow)